# Face Recognition Using the SR-CNN Model

**DOI:** 10.3390/s18124237

**Published:** 2018-12-03

**Authors:** Yu-Xin Yang, Chang Wen, Kai Xie, Fang-Qing Wen, Guan-Qun Sheng, Xin-Gong Tang

**Affiliations:** 1School of Computer Science, Yangtze University, Jingzhou 434023, China; 201603485@yangtzeu.edu.cn; 2National Demonstration Center for Experimental Electrical and Electronic Education, Yangtze University, Jingzhou 434023, China; 500646@yangtzeu.edu.cn (K.X.); wenfangqing@yangtzeu.edu.cn (F.-Q.W.); slsgq@yangtzeu.edu.cn (G.-Q.S.); 3School of Electronic and Information, Yangtze University, Jingzhou 434023, China; 4Key Laboratory of Exploration Technologies for Oil and Gas Resources, Yangtze University, Jingzhou 434023, China; tangxg@yangtzeu.edu.cn

**Keywords:** face matching, scale-invariant feature transform (SIFT), rotation-invariant texture feature (RITF), convolution neural network (CNN), graphic processing unit (GPU), parallel computing

## Abstract

In order to solve the problem of face recognition in complex environments being vulnerable to illumination change, object rotation, occlusion, and so on, which leads to the imprecision of target position, a face recognition algorithm with multi-feature fusion is proposed. This study presents a new robust face-matching method named SR-CNN, combining the rotation-invariant texture feature (RITF) vector, the scale-invariant feature transform (SIFT) vector, and the convolution neural network (CNN). Furthermore, a graphics processing unit (GPU) is used to parallelize the model for an optimal computational performance. The Labeled Faces in the Wild (LFW) database and self-collection face database were selected for experiments. It turns out that the true positive rate is improved by 10.97–13.24% and the acceleration ratio (the ratio between central processing unit (CPU) operation time and GPU time) is 5–6 times for the LFW face database. For the self-collection, the true positive rate increased by 12.65–15.31%, and the acceleration ratio improved by a factor of 6–7.

## 1. Introduction

Face matching is a biometric technology that is widely used in a variety of areas [1], such as public security control, intelligent video monitoring, verification of identity, robot vision, etc. [2]. In the geometrical domain, it is a novel way of converting what the human eyes normally do in recognizing one person from another by implicitly extracting some morphological features [3,4]. It was suggested that multimodal emotion recognition can be combined with speech and facial expressions for emotion analysis [5]. Face matching is the most crucial stage in face recognition. Among these stages, local features are invariant to image scale, rotation, illumination, and angle of view, and still have good matching performance under the influence of noise, occlusion, and other factors. The extraction method of local features mainly includes two steps: detection of feature point and generation of feature description. Compared with the selection of the detection algorithm, the algorithm of local feature description has a more significant impact on the performance of the local feature. Furthermore, how to construct a feature descriptor with robustness and uniqueness is of paramount importance in the research of local features.

The purpose of constructing the feature descriptor is to establish the description of the neighborhood information of the feature points, thereby constructing the descriptive vectors of the feature points. These descriptors are based either on the regional distribution, the spatial frequency, differences, or other features. Mikolajczyk et al. [6] made a systematic analysis and comparison of various descriptors. The experimental results showed that the scale-invariant feature transform (SIFT) algorithm based on gradient distribution and the gradient location and orientation histogram (GLOH) are obviously superior to other descriptors. Therefore, most feature descriptors are built on the bases of distribution of local area information (including gradient, gray, and so on) [7,8].

The convolutional neural network (CNN) demonstrated its high efficiency in image classification [9], object detection [10], and voiceprint recognition [11]. Through deep architecture, it can learn abstract expressions close to human cognition. Recent studies showed that CNNs trained on large and diverse datasets like ImageNet [12] can be used to solve tasks for which they are yet to receive training. Many works adopted the solution of ready-made features based on pre-trained CNN extraction, which achieved good results in popular benchmark tests [13,14,15,16].

Farfade et al. proposed a fully convolutional neural network to detect faces in a wide range of orientations [17]. Reference [18] applied the faster region-based CNN (R-CNN), a state-of-the-art generic object detector, and achieved promising results. Reference [19] used fully convolutional networks (FCN) to generate heat maps of facial parts, and then used the heat map to generate face proposals. Reference [20] used a unified end-to-end FCN framework to directly predict bounding boxes and object class confidences. These methods, however, are relatively slow even on a high-end graphics processing unit (GPU). In References [21,22], faster R-CNN was used for face detection. Reference [23] improve the faster R-CNN framework by combining a number of strategies, including feature concatenation, hard negative mining, multi-scale training, model pre-training, and proper calibration of key parameters.

Proposed by Lowe, the SIFT feature [24,25] is one of the most typical feature descriptors, where it generates a 128-dimensional eigenvector by constructing a three-dimensional gradient histogram of the neighborhood of feature points. The SIFT feature has strong adaptability to scale variation, change of illumination, image transformation, and noise, thus having a better fault tolerance in face matching. However, there are still many areas that need to be improved, and relevant problems attracted the attention of the scholars at home and abroad. Susan et al. [26] applied the fuzzy matching factor to the SIFT model, to form an efficient descriptor of fuzzy filters. Park et al. [27] used a combination of Gabor local binary pattern (LBP) histogram and SIFT-based feature points. Narang et al. [28] proposed facial recognition methods based on SIFT and a classification of a Levenberg–Marquardt Back Propagation (LMBP) neural network. Zhang et al. [29] extracted SIFT features from each image. The feature matrix consisted of extracted SIFT feature vectors sent to CNN for learning.

The above scholars effectively optimized the application of the SIFT model in face recognition. Nonetheless, the traditional SIFT model can easily lead to mismatches when the image has multiple similar regions. When the target image is a grayscale image, CNN may not be as effective as SIFT, because SIFT calculates on the basis of the grayscale image without resorting to color information. The same is true when the color of an object changes dramatically [30]. Vijay et al. [31] showed that neither approach was consistently better than the other, and retrieval gains could be obtained by combining both features.

The CNN network is widely used in face recognition technology due to its good recognition performance. To date, few people retrained the entire CNN network, and the widely used alternative is to fine-tune the CNN network, which already uses a lot of pre-training of tag images. Network fine-tuning is a simple learning method for migration, and data in the target data set are usually selected for training [32].

To this end, we propose a robust method of facial feature matching using an SR-CNN model, which merges the rotation-invariant texture feature (RITF) eigenvector and SIFT eigenvector with CNN features, which effectively improves the effectiveness of feature matching. Moreover, the introduction of a GPU powerfully shortens the computing time of this method, making it more efficient and accurate.

In this paper, we combined the local features of the original image with the global image representation learnt by CNN. A high precision of face matching can be obtained by integrating low-level and high-level features [33,34].

The contributions of this paper are summarized below.

We introduce a deep face recognition model, SR-CNN, which combines multiple features. It can effectively identify a wide range of tasks, while being more robust and discriminative. We propose a SIFT/RITF model. This model improves the rotation invariance of the traditional SIFT model while maintaining the advantages of the original SIFT and RITF models.

We investigate a new CNN model, where the batch normalization layer was introduced, and the parametric rectified linear unit (PReLU) was chosen as the activation function; the network parameters were adjusted using the cross-entropy cost function. The validity of the proposed method was verified by testing on two datasets.

The remainder of this paper is organized as follows: Section 2 proposes the SR-CNN model and illustrates it in five steps. In Section 3, the proposed parallel optimization architecture of the SR-CNN model is introduced. In Section 4, the experimental results are presented, and an analytical discussion is conducted. The conclusions are given in Section 5.

## 2. SR-CNN Model

### 2.1. Framework of the Proposed Method

The traditional SIFT model is a method of extracting the local features of an image. Because only the local feature descriptor of the image is used in the feature matching process, it is inevitable that mismatches will result from the lack of texture and the small size of the target. Due to the similarity of local features, the ability to eliminate these misplaced point pairs is limited [35,36]. The SIFT simply describes the local gradient distribution, which leads to potential mismatches. Some studies use spatial constrains to test and verify SIFT matches [37,38]. These methods reduce the impact of mismatching while introducing more complexity. When there are multiple similar regions of texture information in the face image, only the rotation-invariant texture features of the image region around the key points can achieve good results. In most instances, CNN needs a great deal of training data. The large datasets and cheap computing power provided by GPUs increased CNN’s popularity. Although the results provided by SIFT and other handcrafted methods are not as accurate as those provided by CNN [39,40], they do not need a large number of datasets. However, the modeling ability of the manual approaches is limited by fixed filters, which are still the same for different data sources. Therefore, a novel SR-CNN model with two parts is proposed. The first part involves the extraction of SIFT features and RITF features, which consists of four steps [41,42,43,44]. Firstly, the Gaussian pyramid and the difference of the Gaussian pyramid (DOG) were established, and the space extremum was detected. Secondly, the Taylor formula was adopted to determine the location of the extreme point through five iterations. Secondly, for the key points detected in the DOG, the gradient and distribution of direction of the pixels in the 3σ domain of the pyramid were collected. Finally, the SIFT/RITF descriptors were constructed. The second part involved the CNN firstly using a large sample for pre-training and then using the pre-trained CNN model to extract features. The input of the CNN included the face images after scale normalization, where the images were processed by convolution to extract image features. Then, the results of the convolution layer were processed with the normalization layer to prevent network overfitting. After several convolutional layers and pooling layers, the feature was processed using the fully connected layer, and the result of the CNN feature representation was obtained. The two normalized features were connected together to get the fusion eigenvectors, and the random forest method was used to construct a feature classifier for feature classification. The schematic diagram of the SR-CNN algorithm is shown in Figure 1.

### 2.2. Feature Extraction with SIFT/RITF Model

#### 2.2.1. Detection of Scale-Space Extrema

Firstly, the Gaussian pyramid and the difference of Gaussian pyramid were established, and then the space extremum was detected. In the construction process of the Gaussian pyramid, the size of the image was doubled; on this basis, Gaussian blur was carried out on the image under this size, and the set of a few blurred images constituted an octave. Then, down-sampling was carried out on the most blurred image under this octave, the length and width of which were halved once each, making the area of the image into a quarter of the original. The processed image was the initial image of the next octave, based on the completion of Gaussian blur processing on this octave; this process was repeated until the complete construction of all octaves, thus forming the Gaussian pyramid. Then, the two adjacent images of the same octave were subtracted to obtain the interpolation image. Finally, the DOG pyramid was built using the collection of these images of all octaves.
(1)$$\begin{array}{ll} D(x,y,\sigma ) & =[G(x,y,k\sigma )-G(x,y,\sigma )]⊗I(x,y)\\ & =L(x,y,k\sigma )-L(x,y,\sigma ) \end{array}$$
where *k* is the scale of the Gaussian kernel; the initial value of *k* is set to 1, and the scale is incremented by *k* times. The establishment of the DOG Pyramid is shown in Figure 2.

For each octave of scale space, the initial image was repeatedly convolved with Gauss to obtain the image set shown on the left. By subtracting the Gaussian filter, the difference graph of the Gaussian function on the right could be obtained. After each octave, the Gaussian image was sampled and the process was repeated.

The next step was to traverse and detect the extremum points in three-dimensional space (two-dimensional image, one-dimensional scale). The specific process was to compare the gray value of the current feature point with the other 26 points (eight points adjacent to this point at the current scale, and the nearest 9 × 2 points before and after the scale).

#### 2.2.2. Extrema Localization of Key Points

Since the image is a discrete space, the coordinates of the feature points are all integers, but the coordinates of the key point detected in the scale space are not always integers. To precisely locate the coordinates of key points, the Taylor formula was adopted, and the location of the extreme points was determined through five iterations.
(2)$$D(x)=D+\frac{\partial D^{T}}{\partial x}x+\frac{1}{2}x^{T}\frac{\partial^{2}D}{\partial x^{2}}x.$$

The known point A was used to estimate the value of the nearby point B in the above formula, where *x* = (*x*, *y*, *σ*)*^T^* is the offset of point B relative to point A. The extreme value of Equation (2) was determined by taking the derivative of this formula, and setting it to 0, as follows:(3)$$x_{0}=-\frac{\partial^{2}D^{-1}}{\partial x^{2}}\frac{\partial D}{\partial x}.$$

If the offset is larger than 0.5 in any dimension, indicating that the point is closer to another sample point C, then it is updated as C; otherwise, the iteration ends and *x*_0_ is appended to the current integer coordinates of the feature point. This can be obtained by substituting Equation (3) into Equation (2), as follows:(4)$$D(x_{0})=D+\frac{1}{2}\frac{\partial D}{\partial x}x_{0}.$$

When *D*(*x*_0_) ≤ 0.03 [25], we can determine that the point is with low contrast, which is susceptible to noise and becomes unstable; thus, it needs to be removed.

After precise positioning, there are some edge points in the feature points which are unstable due to some noise; thus, they need to be removed. The characteristic of edge points is that the principle curvature in the direction of the vertical edge is larger, and is smaller in the direction of the tangent to the edge. According to these characteristics, the edge points can be removed. The principle curvature can be obtained using the second-order Hessian matrix, as follows:(5)$$H=\left[\begin{array}{cc} D_{xy} & D_{xy}\\ D_{xy} & D_{yy} \end{array}\right],$$
where *D_xy_* and *D_yy_* can be obtained by subtracting the current pixel points from the surrounding pixel points.

The principal curvature of the point is proportional to the eigenvalue of the Hessian matrices of a certain point; thus, it can be determined by calculating the eigenvalues of the Hessian matrix. Let *α* and *β* be the matrix eigenvalues (assuming *α* > *β*), giving the following equations:(6)$$Tr(H)=D_{xx}+D_{xy}=\alpha +\beta ,$$
(7)$$Det(H)=D_{xx}D_{yy}-{\left(D_{xy}\right)}^{2}=\alpha \beta ,$$
where *Det*(*H*) is the determinant of the matrix, and *Tr*(*H*) is the trace of the matrix. Upon calculating *α*/*β* to compare the principal curvature of the feature point, if *α*/*β* >10, the point is removed.

#### 2.2.3. Orientation Assignment of the Feature Point

For the key points detected in the DOG, the gradient and distribution of direction of the pixels in the 3*σ* domain of the pyramid are collected.
(8)$$\begin{array}{ll} l(x,y)= & \{{\left[F(x+1,y)-F(x-1,y)\right]}^{2}+\\ & {\left[F(x,y+1)-F(x,y-1)\right]}^{2}{\}}^{1/2} \end{array}$$
(9)$$\rho (x,y)=\tan^{-1}\frac{F(x,y+1)-F(x,y-1)}{F(x+1,y)-F(x-1,y)},$$
where *l(x*, *y*) and *ρ*(*x*, *y*) are the modulus and direction of the gradient at (*x*, *y*) in the DOG, respectively.

#### 2.2.4. Construction of SIFT/RITF Descriptor

For the traditional SIFT feature descriptor, firstly, the image was selected from the pyramid according to the feature point scale; then, we calculated the gradient direction and the amplitude of each point in the domain scope. Each feature point was composed of 16 (4 × 4) sub-points, of which each sub-point had vector data of the eight directions, generating 128 (4 × 4 × 8) dimensions of the eigenvector. Finally, to avoid the influence of illumination, the length of the eigenvector was normalized, and the eigenvector was denoted as $F_{sift}$.

Then, for the construction of the feature vector of the RITF, with the feature point as the centre and the longest diagonals *D* of the image as the diameter, a circular region was constructed as the field of the feature points. The field was divided into a number of concentric circular regions, and, upon eliminating the Gaussian blur and zeroing operation, it still maintained a good rotation invariance, while also avoiding the computational cost from the rotation of the image.

On the circumference where the key point is the center and *r* (desirable *D*/5, *D*/4, *D*/3, etc.) is the radius, *m*-points are sampled by the same interval. We compared the size of m pixels (denoted as *I*_0_(*x*, *y*), *I*_1_(*x*, *y*), …, *I_m_*_−1_(*x*, *y*)) with the center pixel (denoted as *I_c_*(*x*, *y*)), and the image binary was implemented. That is, if *I_c_*(*x*, *y*) < *I_i_*(*x*, *y*), (*i* = 0, 1, …, *m* − 1), the sampling pixel was set to 0; otherwise, it was set to 1. Therefore, the calculation formula of the central pixel point of the RITF feature (denoted as $RITF_{m,r}^{ri}$) is as follows:(10)$$RITF_{m,r}^{ri}=\min\{ROR(RITF_{m,r},k)|k=0,1,\cdots m-1\},$$
where
(11)$$RITF_{m,r}=\sum_{i=0}^{m-1}F(I_{i}-I_{c})2^{i-1},F(x)=\{\begin{array}{c} 1,x\geq 0\\ 0,x<0 \end{array},$$
where *ROR*(*x*, *k*) indicates the right cycle shift *k* times of *m*-bit binary *x*.

Upon rotating the image area to the reference direction according to the main direction of the key points, and the image area of 8 × 8 was taken as the area to be described. In this area, for each pixel point *I_i_*(*x*, *y*), (*i* = 1, 2, …, 64), the RITF feature $RITF_{m,r}^{ri}$ performed calculations, which was centered on it, and denoted as *ritf_i_* (*i =* 1,2,…,64). The further the *I_i_*(*x*, *y*) distance is from the center point *I_c_*(*x*, *y*), the less description data are provided. Therefore, it is necessary for *ritf_i_* to weight, and the weight coefficient is as follows:(12)$$\lambda_{i}=\frac{\exp\{-[{\left(x_{i}-x_{c}\right)}^{2}+{\left(y_{i}-y_{c}\right)}^{2}]/{\left(2\beta \right)}^{2}\}}{\left(2\pi \beta^{2}\right)},$$
where (*x_i_*, *y_i_*) and (*x_c_*, *y_c_*) are the coordinates of the pixel point *I_i_*(*x*, *y*) and the center point *I_c_*(*x*, *y*), respectively, and *β* is a constant. Then, all the weighted RITF eigenvalues are composed of a one-dimensional eigenvector, which is denoted as *R_i_*, giving the following:(13)$$R_{i}=[\lambda_{1}\times ritf_{1}\;\lambda_{2}\times ritf_{2}\;\cdots \;\lambda_{64}\times ritf_{64}].$$

Finally, in order to eliminate the effect of the changes due to illumination, *R_i_* was normalized as $\frac{R_{i}}{\left\|R_{i}\right\|}\to R_{i}{}^{\prime }$. Thereby, a 64-dimensional eigenvector $F_{ritf}$ was obtained, which is the RITF feature descriptor for the surrounding area of the key point *I_c_*(*x*, *y*).

The SIFT/RITF feature descriptor is defined as follows:(14)$$F_{sr}=\left[\begin{array}{c} \alpha F_{sift}\\ (1-\alpha )F_{ritf} \end{array}\right],$$
where $F_{sift}$ is a 128-dimensional SIFT feature descriptor, $F_{ritf}$ is a 64-dimensional RITF feature descriptor, $F_{sr}$ is a 192-dimensional SIFT/RITF feature descriptor, and α
(α<1) is a relative weighting factor depending on the pixel size and the type of the image. In this paper, according to the experimental optimization, we set α to be 0.6.

### 2.3. Feature Extraction with CNN Model

The CNN model proposed in this paper is from Alexnet. The CNN model consists of five convolution layers, three max-pooling layer, and two fully connected layers. The last layer adopts a Softmax classifier [9]. After all kinds of images reach the optimal classification effect, the parameters of each layer of the convolutional neural network are determined. Global features are extracted according to these parameters, and the features of the third layer are extracted. The CNN firstly uses a large sample for pre-training, and then uses the pre-trained CNN model to extract features. Figure 3 shows the structure of the CNN network.

#### 2.3.1. Batch Normalization Layer

The Alexnet consists of five convolution layers, three max-pooling layers, and three fully connected layers. The input is the scale-normalized image. The training of the deep network is a complex process; as long as the first few layers of the network change slightly, then the changes in the next few layers will be accumulated and magnified. Once the distribution of the input data changes, the layer requires constant adaptation to the new distribution, thereby affecting the training speed of the network.

The stochastic gradient descent (SGD) is a simple and efficient method for the training depth network; however, it has the disadvantage of us having to carefully adjust model parameters such as learning rate, initial values, weight attenuation coefficient, dropout ratio, and so on. The choice of these parameters is crucial to the training result; thus, we spent a lot of time on this work. To this end, we deleted the local response normalization (LRN) layer and introduced the batch normalization (BN) layer [45]. Due to BN having the ability to improve network generalization, we could remove the dropout parameter. In each layer of the network, a normalization layer was inserted, which is a normalization process (normalized to a mean of 0, a variance of 1).

Consider a mini-batch α of size n. Since the normalization is applied to each activation independently, we can take any dimension as an example. We have n values of this activation in the mini-batch, $\alpha =\{x_{1},\ldots ,x_{n}\}$. The elements of each mini-batch are sampled from the same distribution. Let the normalized values be {${\hat{x}}_{1},\ldots ,{\hat{x}}_{n}$}, and their linear transformations be $\beta =\{y_{1},\ldots ,y_{n}\}$. The forward conduction process formula of the batch normalization network layer is shown in Algorithm 1.

**Algorithm 1.** Batch normalization (BN)**Input:** Values of n over a mini-batch: α={x1,…,xn};   Parameters to be learned: η, λ. **Output:**
β={y1,…,yn};   {yi=BNη,λ(xi)}.1. Mini-batch mean: μ=1n∑i=1nxi;2. Mini-batch variance: σ2=1n∑i=1n(xi−μ)2;3. Normalized value: x^i=xi−μσ;4. Scale and shift yi=ηx^i+λ≡BNη,λ(xi).

In this algorithm, BN can increase the speed of training, whereby it is possible to choose a larger initial learning rate, which can accelerate the attenuation such that the “optimal learning rate” is obtained, increasing the speed of adjustment of the learning rate. Moreover, it also alleviates the problem of internal covariate shift.

#### 2.3.2. Activation Function Layer

The main function of the activation function in neural networks is to add some nonlinear factors to the neural network, such that the neural network can better solve more complex problems. Without the activation function, the weights and deviations are only linearly transformed. The activation function makes backpropagation possible because the error gradient of the activation function can be used to adjust weights and deviations. This is not possible without a differentiable nonlinear function. When a number is applied to a deep neural network, it mainly affects the training of the network in the two processes of forward propagation and backward propagation. Figure 4 shows the PReLU activation function.

The PReLU is a variant of ReLU, where the slope of the negative region is determined by data, and is not predefined. It was found that PReLU converges faster than ReLU [46]. PReLU was chosen over ordinary ReLU to solve the dying ReLU problem; its mathematical representation is as follows:(15)$$y_{i}=\max(0,x_{i})+a_{i}\times \min(0,x_{i}),$$
where $a_{i}$ is a learning parameter; when $a_{i}$ is a fixed nonzero decimal, it is equivalent to LeakyReLU; when it is 0, PReLU is equivalent to ReLU. Its functional expressions are as follows:(16)$$f(y_{i})=\{\begin{array}{cc} y_{i}, & if\;y_{i}>0\\ a_{i}y_{i}, & if\;y_{i}\leq 0 \end{array}.$$

#### 2.3.3. Backpropagation (BP) of CNN Networks

BP neural networks have strong nonlinear mapping capabilities, as well as a high ability of self-learning and self-adaptation, the ability to apply learning results to new knowledge, and a certain fault tolerance. Based on the above advantages, we used BP to reverse-adjust the parameters. Since the cross-entropy loss function can be assigned a linear gradient, effectively preventing the gradient from vanishing, it can overcome the problem of slow updating of variance cost function. In this paper, we used the cross-entropy loss function instead of the mean variance loss function, and the last activation function was replaced by the sigmoid function, because the cross-entropy loss function is more compatible with it. The cross-entropy cost function is defined as follows:(17)$$C=-\frac{1}{n}\sum_{i=1}^{n}\left[y\ln a+(1-y)\ln(1-a)\right],$$
where *y* is the expected output, and a is the actual output. The following equations are its derivatives:(18)$$\frac{\partial C}{\partial w_{i}}=\frac{1}{n}\sum_{i=1}^{n}x_{i}(\sigma (z)-y);$$
(19)$$\frac{\partial C}{\partial b}=\frac{1}{n}\sum_{i=1}^{n}\left(\sigma (z)-y\right).$$

The weight update is affected by σ(z)  − y, which is the impact of the error. Therefore, when the error is large, the weight updates quickly, and, when the error is small, the weight updates slowly.

#### 2.3.4. Feature Vector Extraction

The input face image passes through the convolution layer, pooling layer, and fully connected layer. The output of the second fully connected layer is taken as the feature vector extracted by the CNN network, and is denoted as $F_{cnn}$. The fully connected layer works by looking at the output of the previous layer (the activation diagram representing the high-order characteristics) and determining which functions are the most relevant to the particular class. The structure of the fully connected layer is shown in Figure 5.

### 2.4. Feature Fusion and Classification

The SR-CNN descriptors were constructed from 128-dimensional SIFT descriptors, 64-dimensional RITF descriptors, and 4096-dimensional CNN features, which contain a variety of image information and should adopt different matching strategies. For the SIFT local feature vector *S*, Euclidean distance was used as its matching strategy.
(20)$$d_{S}=\left|S_{i}-S_{j}\right|=\sqrt{{\sum_{m}\left(S_{i,m}-S_{j,m}\right)}^{2}},$$
where *S_i_*_,*m*_ and *S_j_*_,*m*_ represent the *m*-th eigenvector of the *i*-th feature point and the *m*-th eigenvector of the *j*-th feature point (*m* ≤ 128) of the SIFT local feature, respectively.

For the RITF local feature vector *R*, the *χ*^2^ statistic was used as the matching strategy.
(21)$$d_{R}=\chi^{2}=\frac{1}{2}\sum_{m}\frac{{\left(R_{u,m}-R_{v,m}\right)}^{2}}{R_{u,m}+R_{v,m}},$$
where *R_u_*_,*m*_ and *R_v_*_,*m*_ represent the *m*-th eigenvector of the *u*-th feature point and the *m*-th eigenvector of the *v*-th feature point (*m* ≤ 64) of the RITF local feature, respectively.

For a given face image, in this paper, we extracted two groups of features, denoted as $F_{sr}$ and $F_{cnn}$. Considering the large differences in the features of different categories, this paper used the mean and variance of two groups of features to normalize the corresponding feature vectors. The mean and variance of each group’s features were obtained in the training dataset. The two normalized features were connected together to get the fusion feature vector, which is expressed as
(22)$$F=\left\{F_{sr},F_{cnn}\right\}.$$

For the fusion feature vector  F extracted from each face image, this paper used the random forest (RF) learning method [47] to construct a classifier for feature classification. The number of decision trees in the random forest method was set to 100; the classification results had high accuracy with a feature screening mechanism.

## 3. Parallel Optimization of SR-CNN Model

### 3.1. Architecture of Parallel Computing

In 2007, NVIDIA released the official development platform Compute Unified Device Architecture (CUDA) for GPU programming [48]. The core idea of CUDA is based on three important abstract concepts: shared memory, shielding synchronization, and thread group hierarchy.

The GPU can be used for the computing of parallel data, such that many data elements can be executed in parallel with a higher computational density. The model of CUDA programming releases the GPU’s parallel capabilities. Due to the unique hardware architectural design of GPU, it has strong parallel data computing capabilities (a large number of transistors on GPU are used for arithmetic logic operations). Since several data elements run independently on a thread and have high computational density, there is no need for larger data caches.

### 3.2. Design of GPU Implementation

This paper introduces the optimization and acceleration of the SR-CNN model using the aspects delineated in Figure 6. In this framework, the left side of the dotted line indicates that the program is running on the CPU side, and the right side indicates that it is running on the GPU side.

#### 3.2.1. Establishment of DOG and Detection of Extrema

In the process of the establishment of the Gaussian pyramid, images of each layer need to perform Gaussian filtering, which is time-consuming; thus, two-dimensional convolution was used for the calculations. According to their separability, two dimensions (rows and columns) were used to calculate the parallel convolution kernel function.
(23)$$I_{column}(x,y)=I(x,y)\times g_{x}(t);$$
(24)$$I_{row}(x,y)=I_{column}(x,y)\times g_{y}(t)=I(x,y)\times g_{x,y}(t).$$

Since the column convolution is similar to the row convolution, the parallel construction of the Gaussian pyramid is described below in detail using the column convolution operation as an example. The block was set to two dimensions and the thread was set to one dimension. Part of the image data read in blocks after each calculation are stored in shared memory, such that the data communication between threads can speed up the parallel computing of the program. After the establishment of the Gaussian pyramid, the DOG was obtained by directly subtracting the pixels on the image of the adjacent scale. During the subtraction, non-adjacent images were not affected by the calculation, and they could be processed in parallel on GPU. In the same way, it was possible to detect the extreme points in parallel to the pixels of each layer of images in DOG and find the potential key points. Then, the location and scale of the key points were marked using the binary bitmap, which was inserted into the sequence and uploaded to the host terminal.

#### 3.2.2. Histogram Statistics

In this paper, a circular area with a radius of 15 pixels was used as domain information for the key point descriptors. To compensate for the feature point instability caused by no affine invariance, the gradient amplitude was firstly weighted by the Gaussian smoothing in the whole circular domain; then, division of the region was carried on to this field, dividing 2π into 36 equal points, with each region corresponding to a block. The histogram of the feature points could be statistically acquired using gradient information of the pixel gradient in every block, and the value of the maximum group distance (bin) in the histogram was the main direction of the feature point. The data were imported into red/green/blue/alpha (RGBA) texture storage, and the multi-objective rendering algorithm on the advanced graphics card was used to accelerate the processing, to further optimize the calculation. Finally, the calculated results were compressed to a one-dimensional array and passed back to the host terminal from the device terminal, followed by updating the gradient direction on the CPU and establishing the SIFT descriptor.

#### 3.2.3. Construction of RITF Descriptor

Since an image typically has hundreds of feature points, and the calculation of RITF features of each feature point is time-consuming, we processed the RITF on the GPU. Firstly, the images and the data of the feature points were copied from the host to the device, and then saved to global memory; then, each block was assigned to calculate the RITF eigenvalue of a feature point. As a grid can allow up to 65,535 × 65,535 blocks, the number of blocks is much larger than the number of feature points; thus, there is no need to worry about the problem of data overflow. From Equations (10), (12), and (13), we know that the calculations of the RITF eigenvalue and of the weighting coefficient do not affect each other. Therefore, different kernel functions were designed to be called in parallel in two streams, and the RITF eigenvalues and weighting coefficients were stored in different arrays. To ensure that their values corresponded to each other, the calculation order of the RITF eigenvalue was made to be consistent with that of the weighting coefficient. Finally, before the construction of the RITF descriptor, it was ensured that all calculations proceeded. To do this, in these kernel functions, in addition to returning the calculation results, the state information of calculations was also returned.

#### 3.2.4. Feature Matching of SR-CNN

In the process of feature matching, the face image data were uploaded from the host to the device, and we used the GPU to process in parallel. SIFT/RITF feature vectors of each feature point were put in a block of shared memory, and each block was assigned 192 threads; the first 128 threads executed the calculation of Euclidean distance, and the following 64 threads executed the calculation of the *χ*^2^ statistic. For CNN features, each feature map was mapped to a thread block, and each neuron on the feature map was mapped to each thread on the thread block. The three dimensions of the thread grid (*x*, *y*, *z*) correspond to the width, height, and number of each layer, respectively. The settings in the kernel function were as follows: kernel <*z*, *x*, *y*>. The kernel function started *z* thread blocks, each containing x  × y threads, and started z  ×  x  × y threads altogether. Because the number of threads in a thread block was 512, multiple thread blocks were allocated here. Finally, by calling the _syncthreads() function to achieve thread synchronization, each thread was evaluated before executing the subsequent process.

## 4. Experimental Results and Analysis

This section evaluates the performance of the SR-CNN model using a public dataset and a self-collecting dataset. One open dataset was the Labeled Faces in the Wild (LFW) dataset, which is explained in detail later. To verify the validity of the model, we compared the experimental results with other methods. The specific flow is shown in Figure 7.

### 4.1. Experimental Settings

#### 4.1.1. Experimental Database

The LFW dataset contains 13,000 images of human faces collected from the network, each named after the person being photographed. Of these images, 1680 people had two or more different photographs. We used these images for training and testing, and they were generated randomly and independently of the splits for 10-fold cross-validation. All images were 250 × 250 pixels. Figure 8 shows some sample images from the LFW database [49].

The self-collecting face database includes orthotopic face, lateral face, and tilted face images, among others. It contains 1002 images of 50 people, with each image tagged. Images were taken with different levels of light intensity, posture, and expression. All images were in color and manually cut to 300 × 300 pixels.

#### 4.1.2. Experimental Platform

The CPU was an Intel Core i5-6300U processor, clocked at 2.40 GHz, with 8 GB of memory. The GPU was a GeForce GTX 1060 M, with 640 CUDA cores, a memory bandwidth of 80.16 GB/s, and a core frequency of 1097 MHz.

The operating system was Microsoft Windows 10, and the software used was the compiler platform for Visual Studio 2010, CUDA Toolkit 8.0, and OpenCV 3.4.

#### 4.1.3. Experimental Evaluation Index

The example was divided into positive and negative classes. If an instance was positive, samples were predicted as positive samples, denoted as TP. If negative examples were predicted as positive examples, this was denoted as FP. Correspondingly, if negative samples were predicted as negative samples, this was denoted as TN, and if positive classes were predicted as negative samples, this was denoted as FN. $T_{CPU}$ represents the serial execution time for the CPU, $T_{GPU}$ represents the parallel execution time for the GPU, $S_{r}$ is the speed ratio, TPR is the true positive rate, FPR is the false positive rate, and ACC is the recognition accuracy rate; the formulas are given below.
(25)$$TPR=\frac{TP}{TP+FN};$$
(26)$$FPR=\frac{FP}{FP+TN};$$
(27)$$ACC=\frac{TP+TN}{TP+FP+TN+FN};$$
(28)$$S_{r}=\frac{T_{CPU}}{T_{GPU}}.$$

### 4.2. Experimental Analysis

In this subsection, we analyze the performance of the CNN network with different settings, including training epochs, activation function, and whether the CNN network was pre-trained or fine-tuned. The specific flow is shown in Figure 9.

#### 4.2.1. Analysis of the Accuracy in Different Training Epochs

The training epochs of the CNN network inevitably affect the accuracy. Figure 10 shows that the accuracy rate increased with the increase in epoch numbers.

When the network reached the convergence state, the accuracy tended to be stable upon increase in epoch number. The larger the value of the epoch was, the easier it would be to converge, but the longer it would take. As we can see from the graph, when the number of epochs was 20, the network state was already converged, and SR-CNN had the best performance on the self-collection database. Therefore, in this paper, the training epoch of the CNN network used was set to 20.

#### 4.2.2. Analysis of the Accuracy of Different Activation Functions

Residuals in the backpropagation (BP) algorithm diminish with the depth of network transmission, which makes the underlying network unable to be trained effectively. Residual attenuation is closely related to the selection of activation function in the network model. A better activation function can inhibit the residuals in the process of network transmission attenuation, and improve the convergence speed of the models. We used ELU, Tanh, ReLU, and PReLU instead of the sigmoid function in the original network to perform the experiments, and we achieved better results. Then, we compared the network performance with different activation functions using a table of comparison. The development of activation functions plays a crucial role in the further application of deep neural networks. We conducted experiments on two benchmark datasets; we chose the LFW dataset and self-collection dataset as test sets. In this paper, the activation function of the CNN network used was the PReLU activation function. The experimental results are shown in Figure 11.

As can be seen from Figure 11, the matching accuracy with the PReLU activation function was higher than that with the other functions. Meanwhile, with the same activation function, the accuracy was higher when using the self-collecting database.

#### 4.2.3. Analysis of Pre-Trained and Fine-Tuned CNN Networks

Generally, convolutional neural networks trained from scratch are prone to problems, and fine-tuning can quickly converge the network to an ideal state, but there is a large difference between the target dataset and the pre-trained dataset. In the recognition task of the target dataset, it is difficult to extract the visual features of the image using the pre-trained CNN model. In order to make the pre-trained CNN parameters more appropriate to the target dataset, we fine-tuned the CNN models based on the target dataset. With the exception of the last layer, the parameters of the pre-training model were used to initialize other layers, and the output layer was initialized by a zero-mean Gaussian distribution with a standard deviation of 0.01. Unlike the pre-training, we assumed that the weights of the pre-trained CNN were relatively good; thus, we set a small learning rate for the CNN weights to be fine-tuned. We did not want to change them too quickly or too much; thus, we kept our learning rate and learning rate attenuation very small.

As can be seen from Table 1, the fine-tuned network helped boost the accuracy, while the performance could be further improved. Interestingly, fully connected layer 7 (FC7) worked best after fine-tuning, unlike the pre-trained CNN. Therefore, the output of the FC7 layer was taken as the feature vector.

### 4.3. Comparative Experiment on Performance

In this subsection, we evaluate the proposed model in this paper in different situations. The comparisons included face matching methods before and after improvement, our model with other matching models, and the GPU before and after optimization. The details are shown in Figure 12.

#### 4.3.1. Comparison of Face Matching Methods Before and After Improvement

The face database used in this experiment was provided by LFW (the size of each face was 250 × 250 pixels) and the self-collection face database (the size of each face was 300 × 300 pixels), including front face, side face, and rotated face images, among others. The method of face matching under the traditional SIFT model is only suitable for face matching with the rotation of the positive face and images with small rotation (tilt) angles, but the matching effect is poor for faces with large rotation angles. The improved method combined RITF and SIFT, which improved matching for faces with large rotation (tilt) angle, on the basis of maintaining the effect of the original matching. In this experiment, the improved face matching method was compared with the traditional SIFT model for the same face database. The experimental results were averaged over 10 experiments.

For the LFW face database, as shown in Figure 13, the SIFT model could match three pairs of feature points, but only two pairs of matching points were correct. The improved SIFT/RITF model could match 10 pairs of feature points, and eight pairs of matching points were correct.

The matching effects of two methods for the LFW face database are shown in Figure 14 (acc represents the recognition accuracy rate, and tpr is the true positive rate). For the improved face matching method, the recognition accuracy rate improved by 10.97–13.24% and the true positive rate improved by 8.50–10.11% (due to the low number of feature points in LFW’s face database, the threshold of matching point pairs *m* was set to 3).

For the self-collection face database, as shown in Figure 15, the SIFT model had 19 pairs of matching points, for which only nine pairs of matching points were correct. The improved SIFT/RITF model had 31 pairs of matching points, and 23 pairs of matching points were correct.

The matching effects of both methods for the self-collection face database are shown in Figure 16 (acc represents recognition accuracy rate, and tpr is the true positive rate). For the improved face matching method, the recognition accuracy rate improved by 12.65–15.31% and the true positive rate improved by 7.49–9.09% (the threshold of matching point pairs *m* was set to 16).

#### 4.3.2. Comparison of Our Model with Other Face Matching Models

The LFW face database and self-collection database were used to experiment with different improved SIFT models. In this experiment, we compared eleven different models: principal component analysis with SIFT (PCA/SIFT) [50], Gabor/SIFT [51], speeded up robust features with SIFT (SURF/SIFT) [52], partial-descriptor SIFT (PDSIFT) [53], person-specific SIFT [54], SIFT/Kepenekci [55], wavelet transform of the SIFT feature [56], hexagonal SIFT (H-SIFT) [57], and SIFT/RITF. For PCA/SIFT, gradient patches were used around the key points instead of the original patches to make the representation robust to changes in lighting and to reduce the changes that PCA needs to model [50]. Reference [51] proposes to combine Gabor and SIFT for face recognition. The SURF/SIFT method uses a SURF detector with a SIFT descriptor to augment the proficiency of contemporary face recognition systems [52]. The proposed PDSIFT method retains key points at large or near-face boundaries to achieve better performance than the original SIFT [53]. The person-specific SIFT model uses the SIFT features of a particular person and a non-statistical matching strategy to solve the face recognition problem in combination with the local and global similarities on the key point group [54]. SIFT/Kepenekci is a SIFT model-based Kepenekci approach [55]. The method proposed in Reference [56] is based on wavelet transform of the SIFT features. H-SIFT takes advantage of hexagonal transformed image pixels and applies processing on a hexagonal coordinate system, rather than using SIFT on the square image coordinates, displaying better performance [57]. 

The experimental results are shown in Table 2.

According to the experimental results, the SR-CNN model presented in this paper obtained the optimal face recognition performance compared with the other ten methods. As we can see from the Table 3, the SIFT/RITF model performed well with the LFW database, and the CNN model performed well with the self-collection database. Correspondingly, SR-CNN performed well with both databases, which proves that the proposed SR-CNN model is very effective for face recognition. We fully considered the complementary information of different characteristics, which can solve the problems that CNN cannot avoid. We fused complementary features, and the results of the experiment show that the fusion was a success.

Based on the above experiments, compared with the traditional method, the facial feature matching method using the SR-CNN model is greatly improved. For the LFW face database, the size of the face image is smaller. Although there are fewer matching points, the optimization effect was better, and the true positive rate could still be increased to 97.32%. For the self-collection database, the size of the face image was larger, and all of them were color images. Therefore, there were more pairs of matching points, and the optimization effect was better, with the true positive rate as high as 97.61%. In addition, for the improved method, there was also an improvement in the number of matching points pairs, such that the true positive rate with both databases could reach about 99%.

#### 4.3.3. Comparison of GPU before and after Optimization

To carry out the comparison experiment of multi-group data, the LFW face database and the self-collection face database were arranged into five groups according to the number of images. In each experiment, the serial computing of the CPU was compared with the parallel computing of the GPU, and the test time of each group was the average of 10 experiments.

According to Figure 17 and Table 4, it can clearly be seen that, with the increase in the number of face images, the calculating time exhibited an upward trend, the acceleration ratio of which was a factor of about 6–7 for the LFW database.

Similarly, as can be seen from Figure 18 and Table 5, for the self-collection database, the acceleration ratio was a factor of about 5–6.

Based on the above experiments, the use of GPU parallel computing could effectively improve the running speed of the program and reduce the execution time. Compared with the self-collection database, the face size in the LFW database was small; thus, calculation time of the feature extraction was lower for each image, the acceleration ratio was relatively large, and the optimization effect was more distinct. When the number of pictures was small, and there was no comprehensive use of resources on the GPU thread, the acceleration was small and the speed was not very high. However, when the number of images gradually increased, all the threads on the GPU were called, and the acceleration ratio of both databases gradually increased. Therefore, GPU parallel computing is optimal for situations where there are many pictures.

## 5. Conclusions and Future Work

This paper introduced a robust face matching method based on an SR-CNN model. It improved the rotation invariance of the traditional SIFT model while maintaining the advantages of the original SIFT model, CNN model, and the RITF model, which tackles the problem caused by the traditional SIFT model when there are multiple similar regions in an image. Furthermore, the parallel acceleration based on the GPU could not only ensure the matching accuracy, but also effectively improve the running efficiency and shorten the running time. However, in the case of complex backgrounds, the accuracy of face recognition methods will be affected, which will be further studied in the future.

## Figures and Tables

**Figure 1 sensors-18-04237-f001:**
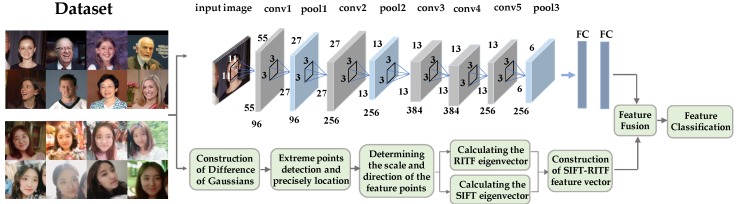
The schematic diagram of SR-CNN model, combining the rotation-invariant texture feature (RITF) vector, the scale-invariant feature transform (SIFT) vector, and the convolution neural network (CNN).

**Figure 2 sensors-18-04237-f002:**
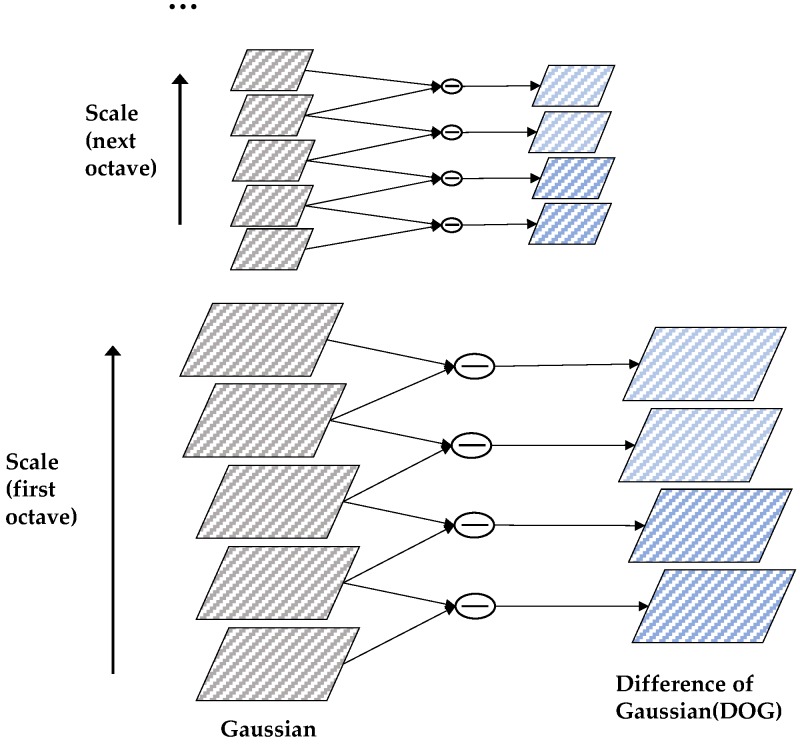
Establishment of the difference of Gaussian (DOG) pyramid.

**Figure 3 sensors-18-04237-f003:**
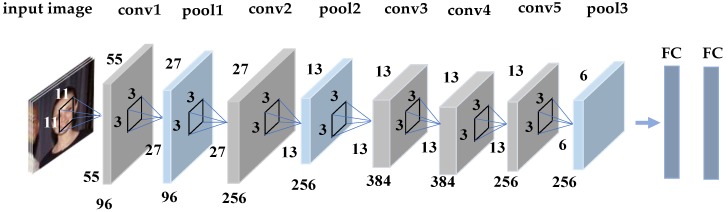
The CNN network structure consists of five convolution layers, three max-pooling layers, and two fully connected layers. The input is the scale-normalized image.

**Figure 4 sensors-18-04237-f004:**
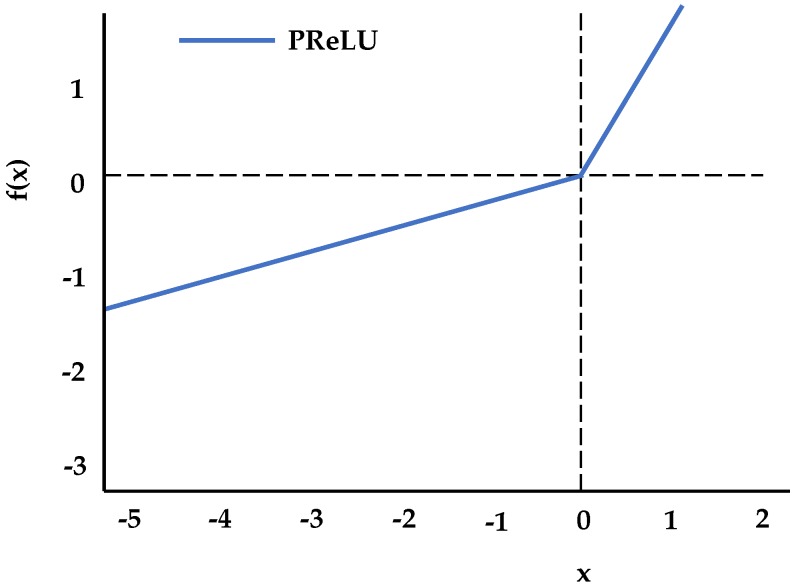
Activation function: parametric rectified linear unit (PReLU).

**Figure 5 sensors-18-04237-f005:**
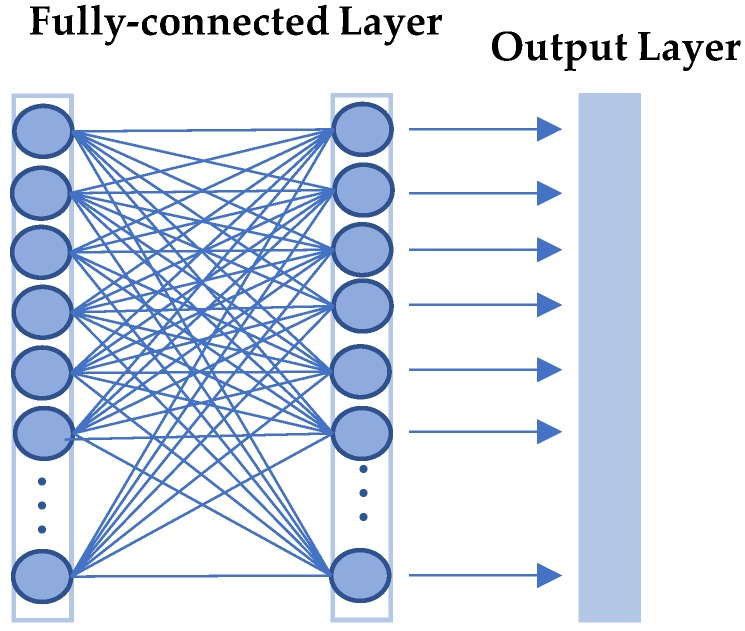
Structure of the fully connected layer.

**Figure 6 sensors-18-04237-f006:**
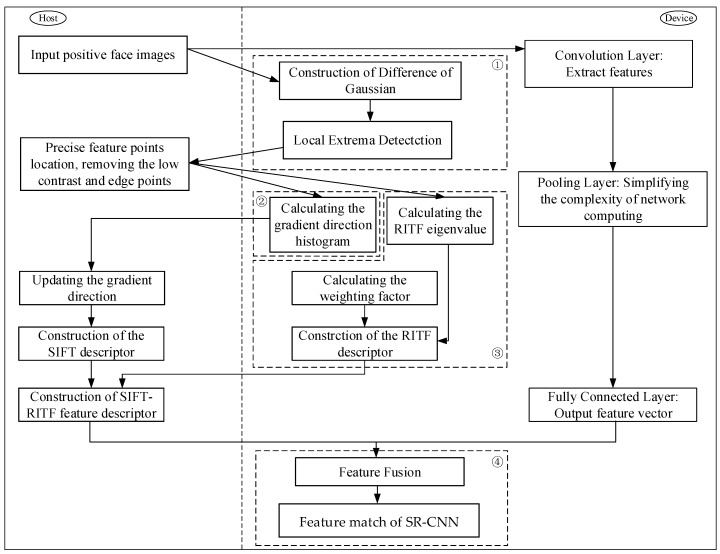
Framework of parallel computing in the SR-CNN model.

**Figure 7 sensors-18-04237-f007:**
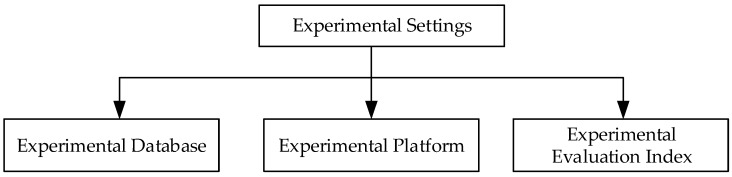
Flow diagram of experimental settings.

**Figure 8 sensors-18-04237-f008:**
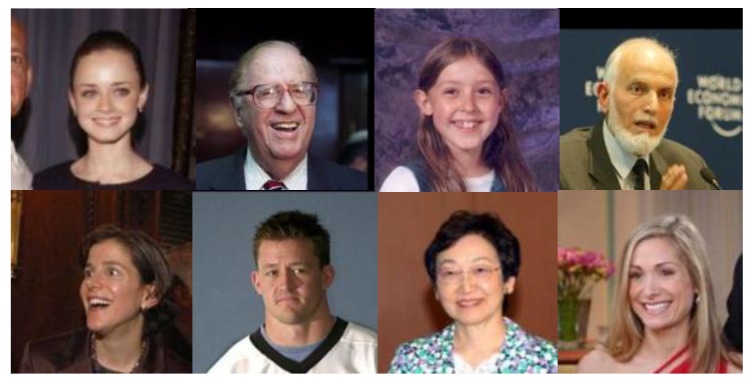
Sample images from the Labeled Faces in the Wild (LFW) dataset.

**Figure 9 sensors-18-04237-f009:**
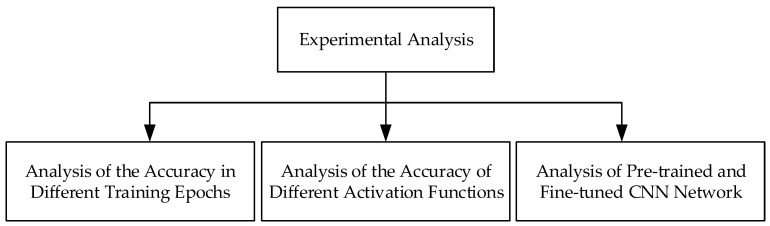
Flow diagram of the experimental analysis.

**Figure 10 sensors-18-04237-f010:**
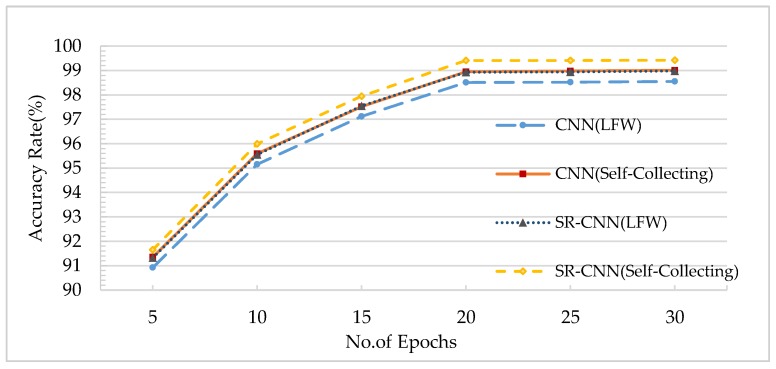
Comparison of accuracy for different training epochs.

**Figure 11 sensors-18-04237-f011:**
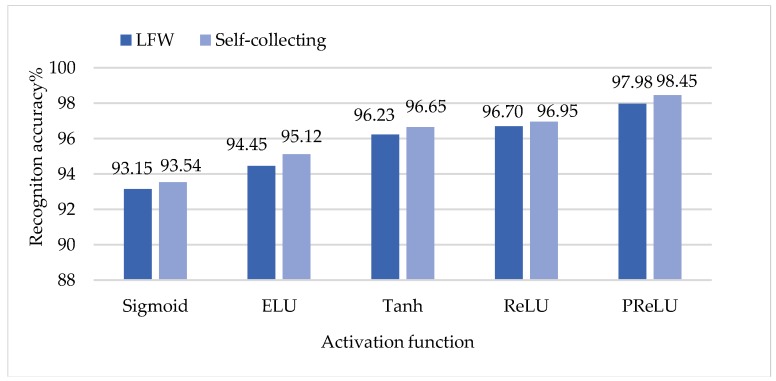
The matching accuracy of the activation function.

**Figure 12 sensors-18-04237-f012:**
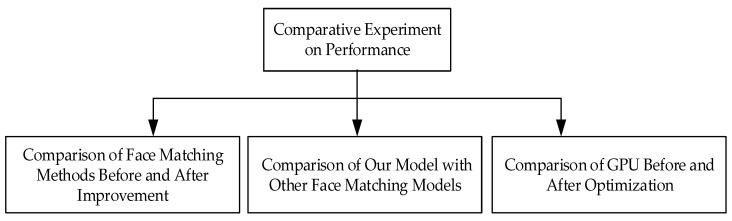
Comparative experiments on performance.

**Figure 13 sensors-18-04237-f013:**
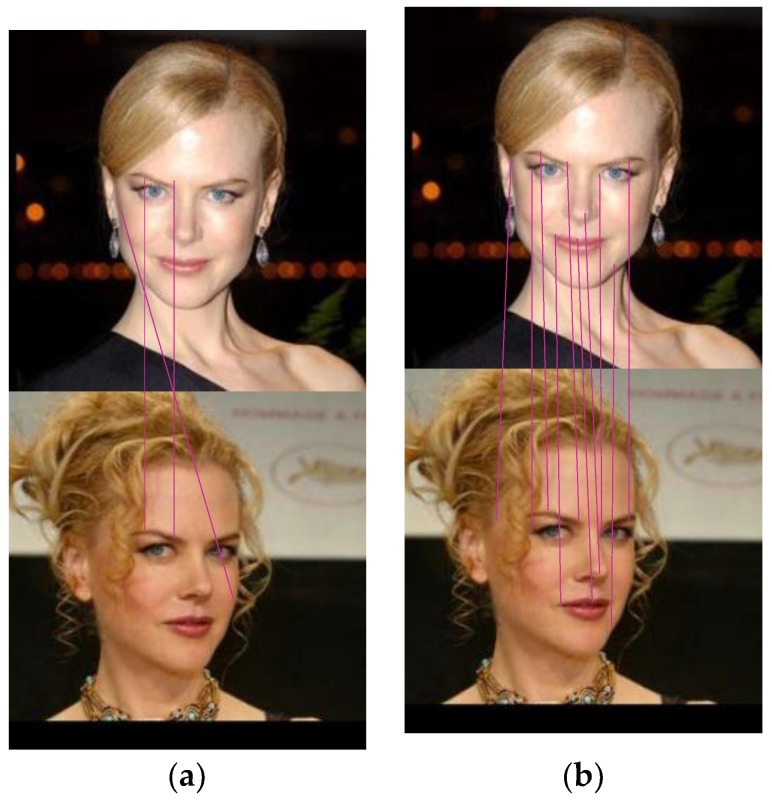
Comparison of matching effect using the LFW face database: (**a**) SIFT Model; (**b**) SIFT/RITF Model.

**Figure 14 sensors-18-04237-f014:**
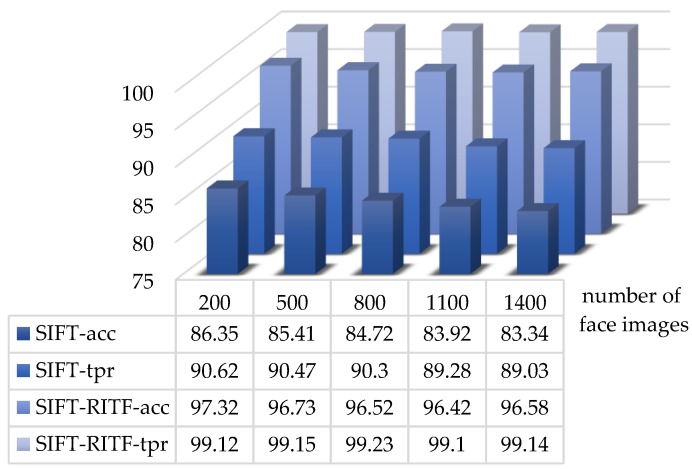
Comparison between SIFT and SIFT/RITF models (LFW face database).

**Figure 15 sensors-18-04237-f015:**
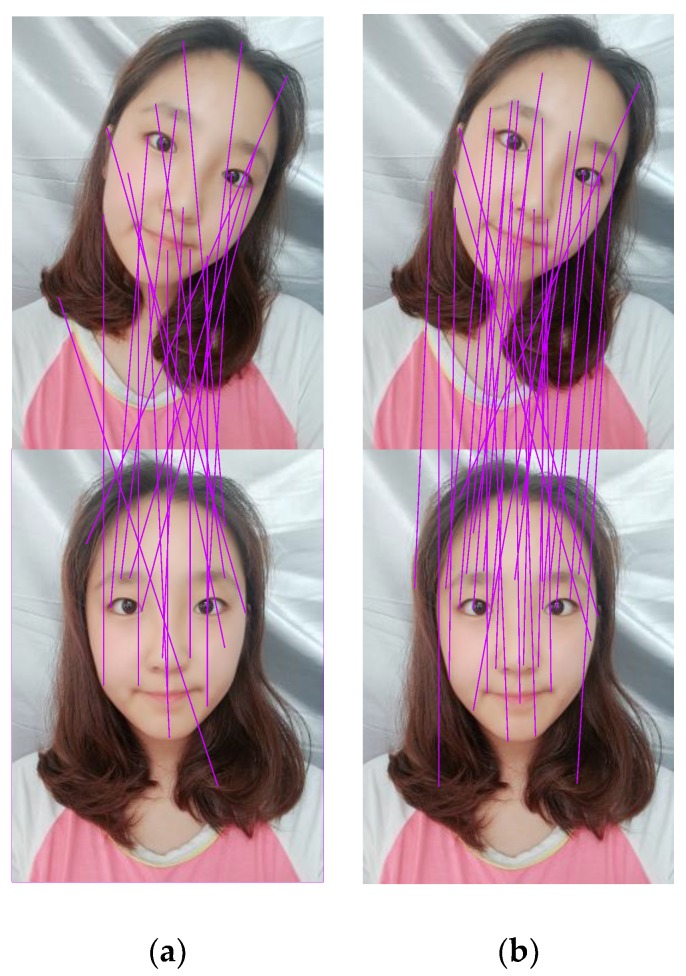
Comparison of matching effect using the self-collection face database: (**a**) SIFT Model; (**b**) SIFT/RITF Model.

**Figure 16 sensors-18-04237-f016:**
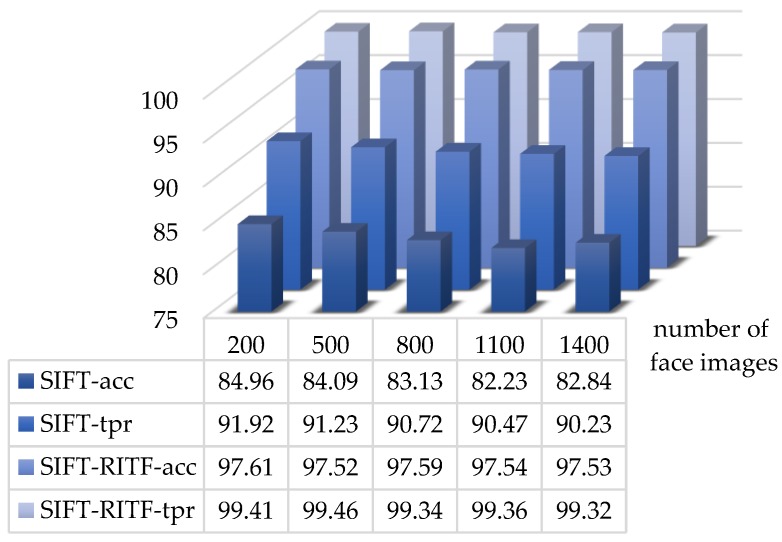
Comparison between SIFT and SIFT/RITF models (self-collection face database).

**Figure 17 sensors-18-04237-f017:**
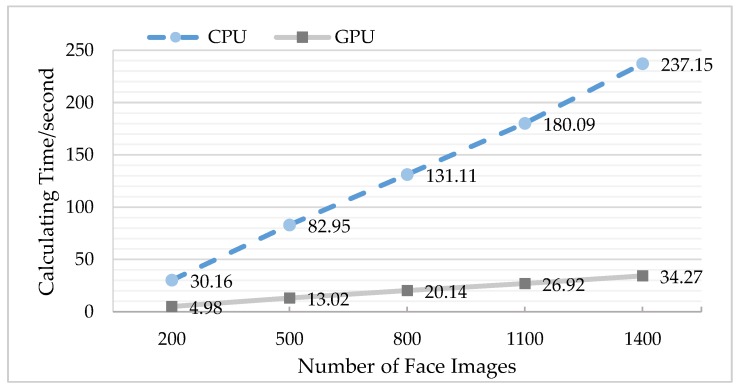
Comparison of central processing unit (CPU) and graphics processing unit (GPU) operation time (using the LFW database).

**Figure 18 sensors-18-04237-f018:**
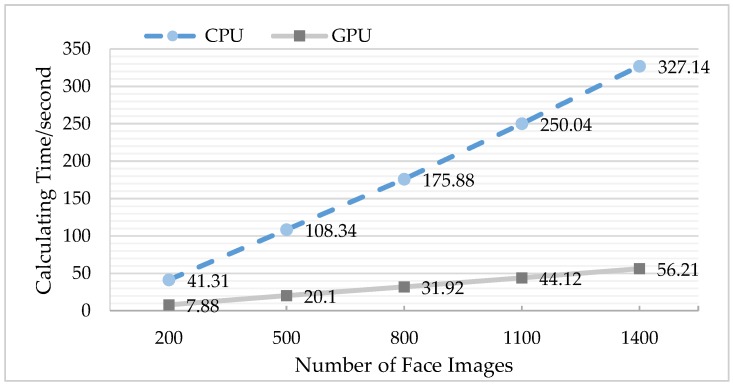
Comparison of CPU and GPU operation time (using self-collection database).

**Table 1 sensors-18-04237-t001:** Comparison between pre-trained convolution neural network (CNN) and fine-tuned CNN for the proposed method. LFW—Labeled Faces in the Wild; Conv—convolution layer; FC—fully connected layer.

Target Datasets	Layer	LFW	Self-Collection
Pre-trained	Conv5	85.3	83.4
	FC6	86.8	85.8
	FC7	86.0	85.4
Fine-tuned	Conv5	86.8	85.8
	FC6	87.7	87.5
	FC7	88.7	88.6

**Table 2 sensors-18-04237-t002:** Recognition performance of different improved the scale-invariant feature transform (SIFT) models using the LFW dataset. ACC—recognition accuracy rate; TPR—true positive rate; FPR—false positive rate; PCA—principal component analysis; SURF—speeded up robust features; PD—partial descriptor; H—hexagonal; RITF—rotation-invariant texture feature; R-CNN—region-based CNN; SR-CNN—CNN with SIFT and RITF.

Method	ACC	TPR@FPR = 1%	TPR@FPR = 0.1%
PCA/SIFT	92.34	89.53	80.46
Gabor/SIFT	94.37	91.76	82.72
SURF/SIFT	95.12	92.35	82.75
PDSIF	95.68	92.65	83.07
Person-specific SIFT	95.72	92.94	83.39
SIFT/Kepenekci	96.35	93.37	83.79
Wavelet transform of the SIFT feature	96.37	93.36	83.51
H-SIFT	96.42	93.44	83.86
SIFT/RITF	98.56	95.51	85.94
CNN	97.98	95.18	85.59
Faster R-CNN	98.45	95.67	85.97
SR-CNN	98.98	95.96	86.47

**Table 3 sensors-18-04237-t003:** Recognition performance of different improved models using the self-collection dataset.

Method	ACC	TPR@FPR = 1%	TPR@FPR = 0.1%
PCA/’SIFT	92.51	89.45	80.52
Gabor/SIFT	93.86	90.81	82.84
SURF/SIFT	94.96	91.45	82.83
PDSIF	95.37	92.74	83.17
Person-specific SIFT	94.84	93.45	83.48
SIFT/Kepenekci	96.12	93.34	83.95
Wavelet transform of the SIFT feature	96.35	93.38	83.76
H-SIFT	96.58	93.56	83.97
SIFT/RITF	98.23	95.67	86.08
CNN	98.87	95.25	85.89
Faster R-CNN	99.10	95.86	86.53
SR-CNN	99.28	96.04	87.01

**Table 4 sensors-18-04237-t004:** Acceleration ratios of central processing unit (CPU) and graphics processing unit (GPU) using the LFW face database.

Number of Face Images	Acceleration Ratio
200	6.06
500	6.37
800	6.51
1100	6.69
1400	6.92

**Table 5 sensors-18-04237-t005:** Acceleration ratios of CPU and GPU using the self-collection face database.

Number of Face Images	Acceleration Ratio
200	5.24
500	5.39
800	5.51
1100	5.66
1400	5.82

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
