# Peer review of "Face Recognition Using the SR-CNN Model"

_sensors, 2018, doi:10.3390/s18124237_

Round 1
Reviewer 1 Report
The paper deals with an interesting topic on face analysis with a specific focus on recognition. In the present form the paper presents many lacks. Starting from the introduction authors should provide a wider picture on the domain in order to provide to the paper a significant scientific level. Some more references should be added as for instance:
Marcolin, F. & al. (2012). Geometrical descriptors for human face morphological analysis and recognition. Robotics and Autonomous Systems, 60(6), 928-939.
Moos, S., ... & al. (2017). Cleft lip pathology diagnosis and foetal landmark extraction via 3D geometrical analysis. International Journal on Interactive Design and Manufacturing (IJIDeM), 11(1), 1-18.
Some more elements should be introduced for what concern the methodology proposed in order to better enphatize the added value of the proposed work.
Many figures are not clear or have graphic problems, quality must be improved
Formulas and matrices must be checked
Author Response
Please see the attached file: Response to Reviewer 1 Comments.pdf.

Reviewer 2 Report
Extraction of local features in images requires accurate feature description. CNN features are not as effective as SIFT for different colors of the same object. This paper proposes a new parallel architecture to process images in a fast manner using RIFT and SIFT features.
The paper appears to be a simple application combining different features. I feel the novelty of the model is low. Here are my specific comments:
Line no 22, why is SIFT useful for scale invariance. CNN is already scale invariant. Similarly in line no 60, they reiterate that SIFT can adapt to illumination and image transformation. All these things are achievable by CNN model alone.
Line no 50, what do you mean by ‘gradient’ ‘gray’ etc.?
Line no 187, is there any reference for this approach?
Line no 223, why is sift feature of 128 dimension. This information should be in experiments.
Line no 584, why is GPU not able to maintain the accuracy?
Comparison with deep learning methods is not there. For example, see ‘Convolutional MKL based multimodal emotion recognition and sentiment analysis’, ICDM, 2016
Author Response
Please see the attached file: Response to Reviewer 2 Comments.pdf.

Reviewer 3 Report
The paper describes a method for face recognition using Rotation-Invariant Texture Feature SIFT feature vector with scale invariance and combined with convolutional neural network implemented on GPU. Results show both good accuracy and processing time.
Please make the following modifications:
-explain more clearly figure 1 and 6 (parallelisation process)
-add reference in text to all figures
-There is no section with related work - add a section that describes existing solution used for evaluation in Section 4.3.2
Author Response
Please see the attached file: Response to Reviewer 3 Comments.pdf.

Round 2
Reviewer 2 Report
The authors have answered my questions satisfactorily.
Author Response

(The authors gave the same response as above.)

Reviewer 3 Report
The authors accessed all my comments. Some minor updates are needed (format editing):
-Line 160: The caption of Figure 2 appears on the same line with the sentence- please add a '\n'
-Line 360: title of the subsection must be placed on the next page
Author Response

(The authors gave the same response as above.)
